# The Association between Symptoms of Nomophobia, Insomnia and Food Addiction among Young Adults: Findings of an Exploratory Cross-Sectional Survey

**DOI:** 10.3390/ijerph18020711

**Published:** 2021-01-15

**Authors:** Haitham Jahrami, Ammar Abdelaziz, Latifa Binsanad, Omar A. Alhaj, Mohammed Buheji, Nicola Luigi Bragazzi, Zahra Saif, Ahmed S. BaHammam, Michael V. Vitiello

**Affiliations:** 1Ministry of Health, Manama, Bahrain; zsaif@health.gov.bh; 2College of Medicine and Medical Sciences, Arabian Gulf University, Manama, Bahrain; sanadlatifa@gmail.com; 3The Walton Centre, Neurology Department, Royal Liverpool University Hospital, NHS, Liverpool L9 7LJ, UK; ammarmyasser@gmail.com; 4Department of Nutrition, Faculty of Pharmacy and Medical Science, University of Petra, Amman 11196, Jordan; omar.alhaj@uop.edu.jo; 5International Inspiration Economy Project, Manama, Bahrain; buhejim@gmail.com; 6Laboratory for Industrial and Applied Mathematics, Departments and Statistics, York University, Toronto, ON M3J 1P3, Canada; 7Department of Medicine, College of Medicine, University Sleep Disorders Center, King Saud University, Box 225503, Riyadh 11324, Saudi Arabia; ashammam2@gmail.com; 8The Strategic Technologies Program of the National Plan for Sciences and Technology and Innovation in the Kingdom of Saudi Arabia, Riyadh 11324, Saudi Arabia; 9Psychiatry & Behavioral Sciences, Gerontology & Geriatric Medicine, and Biobehavioral Nursing, University of Washington, Seattle, WA 98195-6560, USA; vitiello@uw.edu

**Keywords:** behavioral addiction, food addiction, internet addiction, sleep difficulties, sleep problems, smartphone addiction

## Abstract

No previous research has examined the association between symptoms of nomophobia and food addiction. Similarly, only a few studies have examined the association between nomophobia and symptoms of insomnia. This exploratory study utilized an online self-administered, structured questionnaire that included: basic sociodemographic and anthropometrics; the nomophobia questionnaire (NMP-Q); the insomnia severity index (ISI); and the Yale Food Addiction Scale (YFAS) in a convenience sample of young adults (18–35 years) in Bahrain (*n* = 654), 304 (46%) males and 350 (54%) females. Symptoms of severe nomophobia, moderate-severe insomnia, and food addiction were more common among female participants both for each disorder separately and in combination; however, differences did not reach statistical significance. For severe nomophobia, the rate for females was 76 (21.7%) and for males was 57 (18.8%) *p* = 0.9. For moderate-severe insomnia, the rate for females was 56 (16%) and for males was 36 (11.84%) *p* = 0.1. For food addiction, the rate for females was 71 (20.29%) and for males was 53 (17.43%) *p* = 0.3. A statistically significant association was present between nomophobia and insomnia *r* = 0.60, *p* < 0.001. No association was found between nomophobia and food addiction. Nomophobia is very common in young adults, particularly in females; nomophobia is associated with insomnia but not with food addiction.

## 1. Introduction

Mobile devices (smart mobile phones, phablets, and tablets) present great opportunities and comforts to many people [1]. They facilitate productivity in work and school, provide access to entertainment, and help maintain social contacts with others [1,2]. It is unarguable that mobile devices have become an essential everyday component of a modern lifestyle [3]. However, in recent years, a diversity of problems arising from mobile devices have become prominent [4]. Adverse health effects of using mobile devices include repetitive strain injuries [5], muscle tension [6], headaches [7], decreased attention [8], anxiety [9], depression [10], poor concentration [11], sleep problems [12], and change in body weight [13]. Some investigations concluded that the use of mobile devices could lead to addictive, antisocial, and potentially dangerous behaviors [3,14,15]. With the extensive and frequently excessive use of mobile devices and the dependence on the device that can develop, a new pathology known as nomophobia is emerging and being described and defined as a psychiatric disorder [3].

Nomophobia (for “no-mobile-phobia”) has its origin in England [16], and is defined in the Cambridge Dictionary as “the fear or worry at the idea of being without your mobile phone or unable to use it”. The rate of nomophobia among young adults aged between 18–35 years ranges between 75% and 100% in both developed and developing countries [17,18]. The symptoms of nomophobia vary and may include: preoccupation with the mobile device, use in socially inappropriate settings, negative effects on relationships, and development of withdrawal symptoms, e.g., feelings of anger, tension, or depression when the phone is inaccessible [19]. Scholars classify nomophobia as a form of situational phobia, and there have been several calls for its inclusion into the Diagnostic and Statistical Manual of Mental Disorders, Fifth Edition (DSM-5) [14]. People with severe forms of nomophobia have a greater need to constantly check their mobile devices for notifications and status updates [16]. Rising evidence showed that nomophobia is associated with anxiety, anger issues, and decreased productivity and performance [3,10].

At present, there are only a few studies to investigate the association between nomophobia and diets and eating behaviors [20,21]. Existing research concludes that nomophobia is associated with food group consumption [13]. A negative correlation between nomophobia and daily intake of the meat, fish, and eggs group, the vegetable group, and the milk and dairy group was specifically reported [13].

Individuals with high levels of nomophobia are more likely to skip meals [7], have irregular eating habits [22], and consume fewer vegetables, fruits, and dairy products [23,24]. Excessive use of smartphones is related to disordered eating in young adults as indicated by increased consumption of processed and fast food, and increased body mass index [21,25]. Given these relationships it is possible that nomophobia may also be associated with disordered eating/food addiction. Food addiction is a behavioral addiction that is defined by the compulsive consumption of processed foods; it involves either the behavior (eating) or substance (food) [26].

The relationship between nomophobia and sleep problems is better described. A recent systematic review of the nomophobia literature revealed that several studies demonstrated a strong association between nomophobia and sleep problems [3,27,28,29,30]. However, age and sex differences in the association between nomophobia and sleep problems were left unexamined.

Research on nomophobia is in its initial stages, and much remains to be determined. We conducted an exploratory study to examine the relationships among symptoms of nomophobia, insomnia, and disordered eating in the form of food addiction. This exploratory study assessed these three conditions, which have never been examined in a single population, to determine how they might interact as such might justify further research and potentially inform more effective clinical interventions.

## 2. Materials and Methods

### 2.1. Study Design

An exploratory cross-sectional research design was used to examine the associations among symptoms of nomophobia, insomnia, and food addiction among a convenience self-selected sample of young adults (18–35 years) [31] in Bahrain. The Strengthening the Reporting of Observational Studies in Epidemiology (STROBE) guidelines [32] were followed to strengthen the quality of study design and reporting.

### 2.2. Setting and Participants

The study was conducted in August 2020, using self-administered, structured questionnaires. Seven hundred participants who matched the demographic characteristics of the study sample were surveyed using convenience self-selection sampling. The sample was recruited via posting in active instant messaging groups and advertisements on social media platforms, including; WhatsApp, BlackBerry Messenger, Viber, Signal, Line, and social media platforms Facebook, Twitter, and Instagram. Complete data were available for 654 participants and were included in the final analyses. The inclusion criteria were: (1) young adults aged between 18–35 years from both sexes; (2) own at least one mobile device; (3) not dieting or involved in any lifestyle modification program or clinical research; (4) willing to participate in the study. The exclusion criteria were individuals with (1) a history of any psychiatric disorder; and (2) chronic medical conditions. We specifically screened for gastroesophageal reflux disease, diabetes, cardiovascular disease, kidney disease, neurological disorders, respiratory problems, and thyroid disease. The screening process was based on self-reports of a diagnosis made by a physician and/or receiving prescribed regular treatment for the conditions above.

### 2.3. Ethical Consideration

This study was approved by the Secondary Healthcare Research Ethics Committee (REC) of the Ministry of Health; the Kingdom of Bahrain, in June 2020 (SHCRC/EF/14/10/2020). Participation in this study was voluntary, with no monetary or non-monetary incentives given, and the participant was able to withdraw from this study at any time. All procedures performed followed the ethical standards of the 1964 Helsinki declaration and its amendments.

### 2.4. Data Collection

An online self-administered, semi-structured questionnaire was used for data collection. When the participants clicked on the Uniform Resource Locator (URL) link, they were directed to a Google Form containing the questionnaire. Upon completing the questionnaire, the participants were asked to forward the survey link to their social network. All the responses were saved in a protected Google drive and available to be viewed at any time for analysis by the principal investigator. The electronic survey was developed using the Checklist for Reporting Results of Internet E-Surveys” (CHERRIES) [33].

### 2.5. Instruments

The questionnaire consisted of four parts: First, sociodemographic and anthropometric data, which included age, sex, body weight (kilograms), and height (centimeters). Studies have shown that self-reported weight and height are reliable ways to collect anthropometric measurements [34,35]. A recent systematic review and meta-analysis of comparison of self-reported and directly measured weight and height concluded that differences between the two methods are negligible for research and clinical use [36].

Second, the nomophobia questionnaire (NMP-Q) [37] was used to determine the fear of participants for being without a mobile or smartphones. NMP-Q consists of 20-items each scored on a 7-point Likert scale ranging from 1 (“strongly disagree”) to 7 (“strongly agree”). The NMP-Q was originally validated using exploratory factor analysis and has an overall Cronbach’s alpha coefficient of 0.95 [37]. The Arabic validated version of the NMP-Q was used in our research, which has excellent psychometric properties; overall Cronbach’s alpha coefficient was 0.9 [38]. The total score interpretation is as follows: ≤20 Absence of nomophobia, 21–59 mild level of nomophobia; 60–99 moderate level of nomophobia; and 100–140 severe nomophobia.

Third, the insomnia severity index (ISI) [39] was used to determine diurnal and nocturnal insomnia symptoms. The ISI consists of seven items assessing different aspects of insomnia, which include difficulties in initiating or maintaining sleep, early awakenings, satisfaction with sleep quality, interference with functioning, and distress associated with a sleep problem. The total score categories interpretation are as follows: 0–7 = no clinically significant insomnia; 8–14 = subthreshold insomnia; 15–21 = clinical insomnia (moderate severity); and 22–28 = clinical insomnia (severe). the Arabic validated version of the ISI was used which has excellent psychometric properties [40]. The sensitivity and specificity of ISI, 86%, and 88%, respectively [39]. An ISI score of 15 or higher is considered moderate to severe insomnia in clinical populations. A score >10 is considered optimal for detecting insomnia cases in community samples, while ISI scores <7 identify insomnia remission [39].

Fourth, the Yale Food Addiction Scale (YFAS) [41] was developed in 2009 to provide a standardized measure of individuals with symptoms indicative of food addiction. The YFAS includes 25 items and covers the diagnostic criteria for substance dependence according to the DSM-IV. The YFAS includes items that assess specific diagnostic criteria, such as diminished control over consumption, a persistent desire or repeated unsuccessful attempts to quit, withdrawal, and clinically significant impairment. Scoring of YFAS can be presented in two options: (1) a “symptom count” ranging from 0 to 7 that reflects the number of addiction-like criteria endorsed and (2) a dichotomous “diagnosis” that indicates whether a threshold of three or more “symptoms” in addition to if clinically significant impairment or distress has been met. The current study used the dichotomous “diagnosis” algorithm. The YFAS has an excellent psychometric property in clinical, non-clinical, and community samples [42]. The sensitivity and specificity of YFAS are estimated to be 92% and 99.5%, respectively [42]. The Arabic validated version of the YFAS was used, which has excellent psychometric properties [43].

To avoid survey fatigue, the research team avoided over-surveying participants by keeping the survey short and focused, communicated the significance of the research, and pilot tested the content and technical functionality of the web link using 10 volunteers (not included in final analyses). The entire survey questionnaire took about 20 min to complete.

### 2.6. Data Analysis

Before processing, data were checked for missing values and were visualized (QQ plot, Box plot) for potential outliers and normality. Descriptive statistics were used to describe and summarize participants’ sociodemographic variables (age, sex), anthropometric variables (weight, height, BMI), and report the descriptive results of research instruments (NMP-Q, YFAS, ISI). BMI was calculated using metric units by dividing the body weight (kilograms) by the squared height (meters) [44]. Body Mass Index (BMI) was considered as underweight (<18.5 kg/m^2^), normal weight (18.5–24.9 kg/m^2^), overweight (25–29.9 kg/m^2^), and obese (≥30 kg/m^2^) [44].

The arithmetic mean, standard deviation, and if applicable, 95% confidence intervals (95% CI) were reported for continuous data; and frequency count and percentages were used for categorical variables. Independent samples *t*-test and Pearson Chi-square test were used to compare two groups, for continuous and categorical data, respectively.

Pearson product-moment correlation coefficient was used to examine pairwise correlations or data points between the study variables. Multiple regression analysis was used to examine the association between the study variables after adjusting for covariates; coefficients, 95% CI, and *p*-value were reported. Thus, correlation shows the relationship between the two variables, while regression allows us to see how one affects the other [45]. Statistical significance was considered at *p*-value <0.05.

Data analyses were performed using STATA 16.1 [46] and R statistical computing 4.0.2 [47].

## 3. Results

The survey was closed after reaching a total of 700 responses. Forty-six surveys were excluded because of incomplete values. A total of 654 participants, 304 (46%) males, and 350 (54%) females were included in the final analyses. The mean age of the study sample was 27.2 ± 5.1 years. The mean weight for all participants, male participants, and female participants was 68.8 ± 15.1 kg, 77.1 ± 15.3 kg, and 61.6 ± 10.6 kg, respectively. The mean height for all participants, male participants, and female participants was 164.9 ± 8.6 cm, 168.7 ± 9.3 cm, and 161.6 ± 6.3 cm. The mean BMI for all participants, male participants, and female participants were 25.4 ± 5.7 kg/m^2^, 27.4 ± 6.4 kg/m^2^, and 23.7 ± 4.4 kg/m^2^. About half of the participants were overweight or obese. The rate of overweight was similar in both male and female participants, 29.6% and 21.2%, respectively. However, the rate of obesity was greater among men, 34.4% compared to female participants, 9.2%. The result of the sociodemographic and the anthropometrics are presented in Table 1.

Statistical analysis of the results of the NMP-Q, YFAS, and ISI are presented in Table 2. The result showed that the rate of moderate to severe nomophobia is about 93% (95% CI 91–95%). The mean NMP-Q score was 77.0 ± 17.6. Female participants had a slightly higher rate of severe nomophobia compared to male participants; however, the difference was not statistically significant between the two sex groups *X*^2^ (df = 2, *n* = 654) = 0.90, *p* = 0.70. The rate of food addiction for the entire sample was 19% (95% CI 16–22%). Food addiction was slightly more prevalent among female participants, 20.3% compared to male participants, 17.4%; although the difference did not reach statistical significance *X*^2^ (df = 1, *n* = 654) = 0.90, *p* = 0.40. ISI results showed that the rate of moderate-severe insomnia was 14.1% (95% CI 11–17%), with a mean ISI score of 8.2 ± 4.1 for the entire sample. Females more frequently reported having sub-threshold, moderate, and severe insomnia compared to males although these differences were non-significant *X*^2^ (df = 3, *n* = 654) = 6.10, *p* = 0.10 (Table 2).

A total of 119 (18.2%) participants met the criteria for both moderate-severe nomophobia and a ‘diagnosis met’ for food addiction. Females more frequently reported this issue 68 (57%), compared to males 51 (43%) *X*^2^ (df = 1, *n* = 654) = 0.80, *p* = 0.4. A total of 91 (13.9%) participants met the criteria for both moderate-severe nomophobia and moderate-severe insomnia. Cases with this issue were more likely to be females 55 (60%), compared to males 36 (40%) *X*^2^ (df = 1, *n* = 654) = 2.00, *p* = 0.20. For severe nomophobia, the rate for females was 76 (21.7%) and for males was 57 (18.8%). Twenty (3.1%) participants met the criteria for both moderate-severe insomnia and food addiction. Cases with this issue were more frequently in females 15 (75%), compared to males 5 (25%). Nineteen (2.9%) participants had the triple diagnosis of moderate-severe nomophobia, ‘diagnosis met’ for food addiction, and moderate-severe insomnia. These cases were predominantly females 14 (74%), compared to males 5 (26%) *X*^2^ (df = 1, *n* = 654) = 3.20, *p* = 0.07.

Pearson product-moment correlation coefficient for the association between NMP-Q and ISI; NMP-Q and YFAS; and ISI and YFAS were: *r* = 0.60, *p* = 0.001; *r* = 0.01, *p* = 0.84; *r* = −0.01, *p* = 0.66, respectively. The correlation matrix of NMP-Q, ISI, and YFAS is plotted visually using a correlogram Figure 1. The correlation between ISI and NMP-Q is moderate. The correlation between YFAS and ISI or NMP-Q were weak. A series of multiple regression analyses adjusting for age, sex, and BMI revealed that a significant association was only observed between NMP-Q and ISI β = 2.6 (95% CI: 2.3–2.8) *p* < 0.001. Detailed results are presented in Table 3.

## 4. Discussion

Nomophobia is considered as a disorder of contemporary digital and virtual society; and refers to a pathological fear of remaining without mobile technology. Many epidemiological characteristics, risk factors, psychological predictors, behavioral comorbidities, and consequences of nomophobia remain unknown. An improved understanding of nomophobia in various cultural and ethnic groups is needed to avoid hypercodifying modern behaviors as pathological. To that end, the current research provided a novel analysis of nomophobia, food addiction, and insomnia in the same participants.

The main findings of the current research demonstrated that the rate of moderate-severe nomophobia, food addiction, and moderate-severe insomnia symptoms were approximate: 93%, 19%, and 14%, respectively. Nomophobia, food addiction, and insomnia were more common among females than males both as single disorders and in combination. Finally, a statistically significant positive association was present between nomophobia and insomnia.

Our findings are consistent with previous studies reporting that nomophobia is a highly prevalent problem among young adults, especially females, with the majority of the study sample presenting with moderate levels of nomophobia. For example, a multi-center study in Turkey and Pakistan among university students revealed that the rate of nomophobia, according to the NMP-Q, was effectively universal, 99% [18]. The mean NMP-Q scores for Turkey and Pakistan were: 76 and 101, respectively. A recent study in Oman among university students showed that the rate of nomophobia, according to the NMP-Q, was also about 99% with 85% of the sample at a moderate-severe level [17]. The same study found a negative association between nomophobia and academic performance [17]. A mixed-method study in Puducherry, India, on the rate of nomophobia showed that about 80% of college students have moderate-severe nomophobia [27].

Our results demonstrate that symptoms of nomophobia are associated with symptoms of insomnia. This is consistent with several studies, which demonstrated a strong association between nomophobia and sleep problems (mainly insomnia) [3,27,28,29,30,48,49]. A direct relationship was found between the nomophobia score of medical students and sleep quality, measured by the Pittsburgh sleep quality index (PSQI) [50]. Likewise, a one-year prospective study among young Swedish adults revealed increased mobile phone addiction behaviors in individuals with higher baseline scores of sleep problems [51]. A recent study from rural districts in Turkey revealed that the rate of nomophobia was 100%, with 80% at a moderate-severe level [52]. Although it was not formally measured, the same study suggested that excessive use of mobile devices during the day may increase the likelihood of a sleeping disorder [52]. Nomophobia and sleeping problems also appeared to be associated with mobile phone usage (duration per day) among females [53]. There is a wealth of electrophysiological, neuroimmunological, neuroendocrine, and autonomic evidence of arousal and anxiety in people with primary insomnia [54,55,56]. This suggests that anxiety may be a common factor underlying the two conditions (i.e., nomophobia and insomnia). Such is certainly consistent with the hyperarousal model of insomnia and it would seem anxiety would also play a prominent role in nomophobia (almost by definition) [54]. Thus, future studies of nomophobia and sleep problems should include assessments of anxiety.

Other factors might also contribute to the association between mobile device use and insomnia. One such factor is the blue light emitted by smartphone screens interferes with the production of melatonin, the master hormone that regulates circadian rhythm. A recent systematic review concluded that exposure of shortwave blue light of 400–450 nanometer for two hours is sufficient to significantly suppress melatonin [57]. The same review documented that melatonin concentrations start to recover rapidly (after 15 min) from refraining from artificial lights [57]. Experimental findings among healthy young adults revealed that exposure to blue light increases activation of the prefrontal cortex (PFC), especially the dorsolateral PFC (DLPFC) and ventrolateral PFC (VLPFC), which increases working memory and alertness and thus interferes with sleep [58]. Blue light also interferes with some glucocorticoid hormones, including cortisol and some sympathetic nervous system markers such as α-amylase, which both affect stress level and may induce long term insomnia symptoms [59]. The clinical implications of this finding are to screen individuals with chronic insomnia for nomophobia and offer behavioral interventions to reduce the impact of both problems. Nomophobia also has many commonalities with anxiety symptoms (e.g., discomfort, anguish, nervousness, etc.) [27]; and anxiety has a strong association with insomnia symptoms [60]. Further research is needed to explore the interaction between nomophobia, anxiety, and insomnia. Particularly longitudinal studies are needed to detect the development of insomnia or changes in sleep quality of individuals with high symptoms of nomophobia (and anxiety). Additionally, insomnia is a heterogeneous disorder that involves difficulty initiating sleep, maintaining sleep, and/or early morning awakening [61]; thus, future studies involving nomophobia and insomnia need to address the commonality of each component.

Our findings demonstrate that about 19% of our participants meet the diagnosis of food addiction. However, no significant association between food addiction and (a) nomophobia or (b) insomnia was observed. A meta-analysis of twenty-five studies found that the weighted mean rate of YFAS food addiction diagnosis was 19.9% in young adult population samples [62] completely in alignment with the current findings. Moreover, the rate of food addiction was found to be double in an overweight/obese population than those with normal BMI (24.9% and 11.1%, respectively). The current study reported a higher percentage in females (12.2%) than in males (6.4%), and a higher rate in adults older than 35 years old compared to adults younger than 35 years old (22.2% and 17.0% respectively) [62]. Another systemic review focused on papers published in 2014–2017, and reported that more studies were done with female-only samples; however, studies that included both genders found a link between food addiction and female gender [63]. In both reviews, the most common symptoms reported were generally “unsuccessful attempts to cut-down food intake” and a clear association between YFAS scores and Eating Disorders was established [62,63].

The main strengths of this study are: (1) the use of three standardized instruments with excellent psychometric properties to answer the research questions; and (2) the large sample size. However, there are few limitations to the study, which include: (1) data were obtained using self-reports and various types of biases, e.g., recall bias, may become a challenge; (2) the sample is based on a self-selecting convenience sample that is unlikely to be representative of the entire young adult population; (3) because this is a cross-sectional study; causality could not be determined; (4) the limited nature of the questionnaire did not allow for examination of possible contributing factors such as anxiety, depressive symptoms, and others. All questionnaires used in this research are based on symptoms not on a diagnosis. The issue of diagnosis versus symptoms is not only relevant to nomophobia but also insomnia and food addiction. The insomnia questionnaire can provide a catalog of symptoms suggestive of a diagnosis of insomnia but is not the same as a clinical diagnosis of insomnia. Similarly, the food addiction questionnaire can provide an index of symptoms of individuals’ ‘at risk’ of food addiction; however, a clinical diagnosis can be only made via an interview with a qualified healthcare professional.

## 5. Conclusions

Symptoms of nomophobia are extremely common in young adults. The rate of moderate-severe nomophobia, food addiction, and moderate-severe insomnia symptoms were approximate: 93%, 19%, and 14%, respectively. Severe nomophobia, moderate-severe insomnia, and food addiction were more common among female participants both for each disorder separately and in combination. Nomophobia was positively related to insomnia. There was no association between nomophobia and food addiction.

## Figures and Tables

**Figure 1 ijerph-18-00711-f001:**
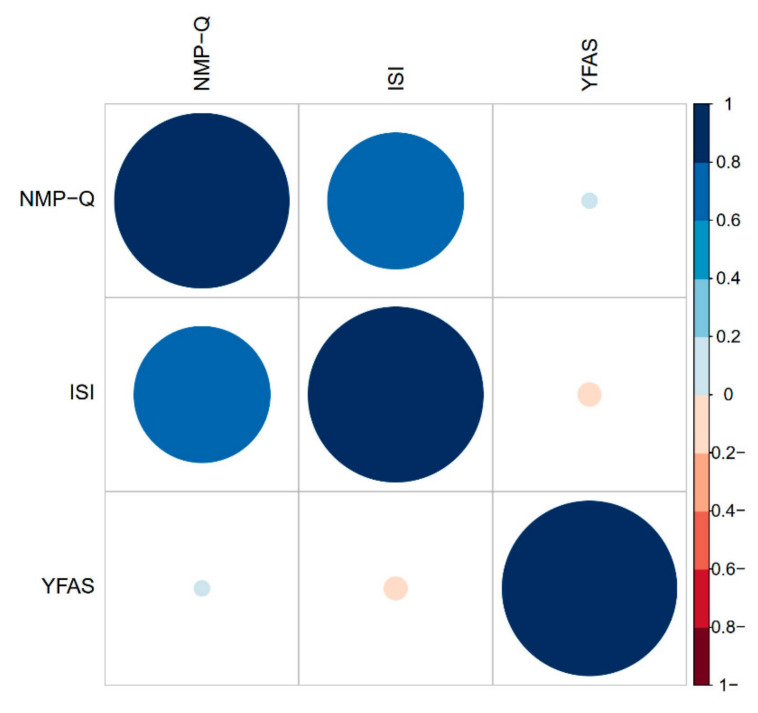
Correlogram of Nomophobia Questionnaire (NMP-Q), Insomnia Severity Index (ISI), and Yale Food Addiction Scale (YFAS).

**Table 1 ijerph-18-00711-t001:** Sociodemographic and anthropometric characteristics of the study participants.

Variable *	All Participants, *n* = 654
Sex	
Male	304 (46%)
Female	350 (54%)
Age (year)	27.2 ± 5.1
Weight (kg)	68.8 ± 15.1
Height (cm)	164.9 ± 8.6
BMI (kg/m^2^)	25.4 ± 5.7
BMI classification	
Underweight	55 (8.6%)
Normal	259 (40.3%)
Overweight	196 (30.5%)
Obese	133 (20.7%)

* Mean ± SD OR Frequency count and (%).

**Table 2 ijerph-18-00711-t002:** Results of study participants Nomophobia Questionnaire (NMP-Q), Insomnia Severity Index (ISI), and Yale Food Addiction Scale (YFAS).

Variable *	All Participants, *n* = 654	Male, *n* = 304	Female, *n* = 350	*X*^2^ or t Statistic	*p*-Value **
Nomophobia Questionnaire (NMP-Q)					
No nomophobia	Nil (0%)	Nil (0%)	Nil (0%)	*X*^2^ (df = 2) = 0.9	0.7
Mild nomophobia	43 (6.6%)	20 (6.6%)	23 (6.6%)		
Moderate nomophobia	478 (73.1%)	227 (74.7%)	251 (71.7%)		
Severe nomophobia	133 (20.3%)	57 (18.8%)	76 (21.7%)		
Yale Food Addiction Scale (YFAS)					
No diagnosis				*X*^2^ (df = 1) = 0.9	0.4
Diagnosis met	530 (81.0%)	251 (82.6)	279 (79.7%)		
	124 (19.0%)	53 (17.4%)	71 (20.3%)		
Insomnia Severity Index (ISI)					
No clinical insomnia	422 (64.5%)	211 (69.4%)	211 (60.3%)	*X*^2^ (df = 3) = 6.1	0.1
Subthreshold insomnia	140 (21.4%)	57 (18.8%)	83 (23.7%)		
Moderate insomnia	83 (12.7%)	33 (10.9%)	50 (14.3%)		
Severe insomnia	9 (1.4%)	3 (1.0%)	6 (1.7%)		
Mean Nomophobia Questionnaire (NMP-Q)	77.0 ± 17.6	76.3 ± 17.1	77.6 ± 18.1	t (df = 652) = 0.9	0.3
Mean symptoms Yale Food Addiction Scale (YFAS)	1.9 ± 0.8	1.9 ± 0.7	1.9 ± 0.8	t (df = 652) = 0.3	0.8
Mean Insomnia Severity Index (ISI)	8.2 ± 4.1	7.9 ± 3.9	8.4 ± 4.3	t (df = 652) = 1.7	0.1

* Frequency count and (%) or Mean ± SD; ** Independent samples *t*-test or Pearson’s Chi.

**Table 3 ijerph-18-00711-t003:** Association between nomophobia (Nomophobia Questionnaire, NMP-Q score), insomnia (Insomnia Severity Index, ISI score), and food addiction (Yale Food Addiction Scale, YFAS score.

Outcome Variable = Nomophobia (Nomophobia Questionnaire, NMP-Q)		
Explanatory Variable	β (95% CI)	*p*-value *
Insomnia (Insomnia Severity Index, ISI)	2.6 (2.3–2.8)	0.001
Food addiction (Yale Food Addiction Scale, YFAS)	0.2 (−1.6–1.9)	0.84
Outcome Variable = Food addiction (Yale Food Addiction Scale, YFAS)		
Explanatory Variable	β (95% CI)	*p*-value *
Insomnia (Insomnia Severity Index, ISI)	2.6 (2.3–2.8)	0.64
Nomophobia (Nomophobia Questionnaire, NMP-Q)	−0.03 (−0.01–0.01)	0.84
Outcome Variable = Insomnia (Insomnia Severity Index, ISI)		
Explanatory Variable	β (95% CI)	*p*-value *
Nomophobia (Nomophobia Questionnaire, NMP-Q)	0.15 (0.13–0.16)	0.001
Food addiction (Yale Food Addiction Scale, YFAS)	−0.1 (−0.5–0.3)	0.64

* Multiple regression adjusting for age, sex, and BMI.

## Data Availability

Available upon request.

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
