# Peer review of "The Association between Symptoms of Nomophobia, Insomnia and Food Addiction among Young Adults: Findings of an Exploratory Cross-Sectional Survey"

_ijerph, 2021, doi:10.3390/ijerph18020711_

Round 1

Reviewer 1 Report

The authors have addressed all of my previous concerns. I have no further comments. Thank you. 

Author Response

The authors have addressed all of my previous concerns. I have no further comments. Thank you.

Response:

We thank the reviewer for taking the time to review our paper, for strengthening it, and for the nice comments.

Reviewer 2 Report

Abstract:

L.37 - Please can the authors add the r-value

Introduction:

L.70-71: "Existing research concludes that nomophobia is associated with food group consumption" - Please can the authors clarify what this relationship - e.g. what is food group consumption in that context?

L.80 : Please add the citations for 'several studies' - comparable to the discussion

Methods:

L.131-132: Please can the authors improve the grammar of the sentence

L.137-137: Why is Mild/Moderate/Severe capitalised?

L.151: Please consider replacing "suspicious" with indicative 

L.173: Please consider replacing "calculated" with a more suitable word

Results:

L.202: Please replace "patients" with paticipants

L.209-2010: Please can the authors improve the grammar of the sentence

L.211-L.222: Please consider replacing "had a dual diagnosis" with "met criteria for both, XX and XY..."

L.225-226: Please can the authors consider restructuring the sentence to improve the clarity and comment on the strengths of the effects, e.g. small/medium/large

Discussion:

L.247: ..."insomnia was considerably low" - also, can the authors comment on how the percentages relate to general findings? E.g. 14% meeting insomnia criteria seem relatively high to me.

L.270-271: Please can the authors expand upon this relationship? I.e. what is meant with mobile phone usage ? Duration?

L.291-292: This should be integrated with the previous point about anxiety/insomnia/nomophobia?

L.2091-298: Not clear why that would be interesting?

Author Response

Response:

We thank the reviewer for taking the time to review our paper, for strengthening it, and for the nice comments.

Abstract:

L.37 - Please can the authors add the r-value

Response:

r-value was added before the p-value in L. 34.

Kindly note that the line numbers changed after the journal formatted the paper.

Introduction:

L.70-71: "Existing research concludes that nomophobia is associated with food group consumption" - Please can the authors clarify what this relationship - e.g. what is food group consumption in that context?

Response:

We have added the following: A negative correlation between nomophobia and daily intake of the meat, fish, and eggs group, the vegetable group, and the milk and dairy group was specifically reported [13].

This was added L. 70-72.

L.80 : Please add the citations for 'several studies' - comparable to the discussion

Response:

We have added citations for several studies from the discussion.

This was added L. 83.

Methods:

L.131-132: Please can the authors improve the grammar of the sentence

Response:

We corrected the sentence and shortened it to: “The questionnaire consisted of four parts:”

L.137-137: Why is Mild/Moderate/Severe capitalised?

Response:

Capitalized was removed from Mild/Moderate/Severe.

Kindly, see L. 145-146.

L.151: Please consider replacing "suspicious" with indicative 

Response:

We thank you for thoroughly reading the manuscript and assisting us in improving it.

We replaced suspicious" with indicative 

Kindly, see L. 160.

L.173: Please consider replacing "calculated" with a more suitable word

Response:

We thank you for thoroughly reading the manuscript and assisting us in improving it.

We replaced "calculated" with considered 

Kindly, see L. 185.

Results:

L.202: Please replace "patients" with participants

Response:

We thank you for thoroughly reading the manuscript and assisting us in improving it.

We replaced "patients" with participants 

Kindly, see L. 217.

L.209-2010: Please can the authors improve the grammar of the sentence

Response:

We thank you for thoroughly reading the manuscript and assisting us in improving it.

We have improved the sentence.

L.211-L.222: Please consider replacing "had a dual diagnosis" with "met criteria for both, XX and XY..."

Response:

We thank you for thoroughly reading the manuscript and assisting us in improving it.

We replaced "had a dual diagnosis" with "met criteria for both

Kindly, see L.228, L.231, L.235

L.225-226: Please can the authors consider restructuring the sentence to improve the clarity and comment on the strengths of the effects, e.g. small/medium/large

Response:

We have revised the sentence for clarity. See now L.243-L.244.

We also explained in Figure 1 “The correlation between ISI and NMP-Q is moderate. The correlation between YFAS and ISI or NMP-Q were weak”. See L. 246-247.

Discussion:

L.247: ..."insomnia was considerably low" - also, can the authors comment on how the percentages relate to general findings? E.g. 14% meeting insomnia criteria seem relatively high to me.

Response:

We agree the sentence was potentially confusing so we re-wrote as follow:

The main findings of the current research demonstrated that the rate of moderate-severe nomophobia, food addiction, and moderate-severe insomnia symptoms were approximate: 93%, 19%, and 14%, respectively.

L.270-271: Please can the authors expand upon this relationship? I.e. what is meant with mobile phone usage? Duration?

Response:

We clarified that mobile phone usage was referring to (duration per day).

L.291-292: This should be integrated with the previous point about anxiety/insomnia/nomophobia?

Response:

We agree, the sentence was integrated the previous point. See L.300-L.301.

L.2091-298: Not clear why that would be interesting?

Response:

Thank you. We have revised the presentation of this point and changed it to a milder suggestion of exploration. Kindly, see L. 319.

Reviewer 3 Report

Dear Authors,

Thank you very much for giving me the opportunity to read your article, which I found challenging and scientifically well-written. Perhaps, in the Introduction you could more expand on the similar studies, which you then also cite in Discussion. In addition, is there any association between nomophobia and physical activities?

Best regards,

Reviewer

Additional comments:

This exploratory cross-sectional survey explores the association between the symptoms of nomophobia, insomnia and food addiction among young adults in Bahrain. The findings indicate that there is no connection between nomophobia and food addiction, however, the symptoms of nomophobia closely associate with the symptoms of insomnia. The topic of the study is novel and brings quite conclusive results.

Author Response

Dear Authors,

Thank you very much for giving me the opportunity to read your article, which I found challenging and scientifically well-written.

Response:

We thank the reviewer for taking the time to review our paper, for strengthening it, and for the nice comments.

Perhaps, in the Introduction you could more expand on the similar studies, which you then also cite in Discussion.

Response:

We have expanded the introduction by adding citations from the discussion. Kindly see L.83. 

In addition, is there any association between nomophobia and physical activities?

Response:

Thank you very much. Physical activities were not part of the scope of our study. 

Best regards,

Reviewer

Additional comments:

This exploratory cross-sectional survey explores the association between the symptoms of nomophobia, insomnia and food addiction among young adults in Bahrain. The findings indicate that there is no connection between nomophobia and food addiction, however, the symptoms of nomophobia closely associate with the symptoms of insomnia. The topic of the study is novel and brings quite conclusive results.

Response:

Thank you so much for your positive encouragement.

This manuscript is a resubmission of an earlier submission. The following is a list of the peer review reports and author responses from that submission.

Round 1

Reviewer 1 Report

Dear Authors,

The Topic of Your Research is very up-to-date, but the scientific methods used in this study are with low quality.

First is the questionnaire for measuring Nomophobia validated?

Second-You write, that the Authors have screened for somatic diagnosis as esophageal reflux, diabetes etc. but there is no describtion of this screening method.

How could BMI been validated?

All conclusions are based on Symptoms, not on any Diagnosis, the Inclusion of the participants is without clear criteria, and the results are diffuse.

Author Response

Dear Authors,

The Topic of Your Research is very up-to-date, but the scientific methods used in this study are with low quality.

Response:

We thank the reviewer for taking the time to review our paper, and for the nice comment about the novelty of the study. We have highlighted the merits and strengths of the scientific methods as per suggestions.

First is the questionnaire for measuring Nomophobia validated?

Response:

We have clarified that – “The NMP-Q was originally validated using exploratory factor analysis technique; and has an overall Cronbach’s alpha coefficient of 0.95”.

Second-You write, that the Authors have screened for somatic diagnosis as esophageal reflux, diabetes etc. but there is no description of this screening method.

Response:

We have clarified that – “The screening process was based on self-reports of a diagnosis made by physician and/or receiving prescribed regular treatment for the list mentioned above.” 

How could BMI been validated?

Response:

We have clarified that – “Studies have shown that self-reported weight and height are reliable ways to collect anthropometric measurements [32,33]. Recent systematic review and meta-analysis of a comparison of self-reported and directly measured weight and height concluded that differences between the two are negligible regarding research and clinical use [34].”

All conclusions are based on Symptoms, not on any Diagnosis, the Inclusion of the participants is without clear criteria, and the results are diffuse.

 Response:

Agreed. We have clarified that – “All questionnaires used in this research are based on symptoms not on a diagnosis; thus, results need to be interpreted from that perspective. The issue of diagnosis versus symptoms is not only relevant to nomophobia but also to insomnia and food addiction. The insomnia questionnaire can provide a catalogue of symptoms suggestive of a diagnosis of insomnia, but is not the same as a clinical diagnosis of insomnia. Similarly, the food addiction questionnaire can provide an index of symptoms of individuals ‘at risk’ of food addiction; however, clinical diagnosis can be only made via an interview with a qualified healthcare professional.” This was added at the end of the discussion before making conclusions.

Reviewer 2 Report

Summary. The presented manuscript reports on a survey conducted in the general young population of Bahrain and investigates the association between Nomophobia and Food addiction; and Nomophobia and Insomnia. Results show that nomophobia is positively correlated with insomnia severity, but not with food addiction.

While the presented article gives some interesting insights, major corrections are proposed to make it suitable for publication.

Major comments:

Introduction. The introduction reads nicely but it doesn’t ‘arrive’ at testable hypotheses at the end. It needs to get clearer why a positive association between insomnia and nomophobia is expected (e.g. underlying pathways and mechanisms) and similarly why a positive association between food addiction is expected (again what are the underlying assumptions of this relationship), same for an expected interaction effect. The introduction needs to convey more clearly why this research is relevant and why this survey was conducted. It also needs to get clearer why there was a focus on the younger population and why there is any advantage of having the three concepts in one survey (same as why there would be an interaction effect)? I would like to ask the authors to rewrite the introduction to improve clarity and focus at the moment the only argument is that no one has done it before – which doesn’t prove the relevance of the question.

Results. In line with standard reporting of statistical results (see Bakker 2011, Behavior research methods) please can I ask the authors to report the statistical values, including t-values, chi-square values, and exact p-values for all comparisons that are the described throughout the manuscript and tables.

Please can the authors explain the reasoning behind conduction person correlations AND multiple regression analysis? It should be clear from the outset which analysis will be used and why. Please can the authors also clearly distinguish between meeting criteria on a questionnaire and an actual diagnosis of a disorder.

Discussion. The discussion reads at parts more like an introduction. Especially P6.L225 – P7.L244 this section lists information that should have already been introduced in the introduction. Furthermore, the discussion is not focused, instead it reads very listwise with future direction that, again, are listed but not expanded upon. It needs to be clear which research studies are needed to investigate the relationships found in the results and why. And it needs more than just a statement saying more research is needed. E.g. what kind of studies are the authors proposing?

I have a few minor corrections to propose. However, they will make more sense once the points above are addressed.

Author Response

Summary. The presented manuscript reports on a survey conducted in the general young population of Bahrain and investigates the association between Nomophobia and Food addiction; and Nomophobia and Insomnia. Results show that nomophobia is positively correlated with insomnia severity, but not with food addiction.

While the presented article gives some interesting insights, major corrections are proposed to make it suitable for publication.

Response:

We thank the reviewer for taking the time to meticulously review our paper, for the constructive feedback to strengthen the report of the study, and for the nice comment about the novelty of the study.

Major comments:

Introduction. The introduction reads nicely but it doesn’t ‘arrive’ at testable hypotheses at the end. It needs to get clearer why a positive association between insomnia and nomophobia is expected (e.g. underlying pathways and mechanisms) and similarly why a positive association between food addiction is expected (again what are the underlying assumptions of this relationship), same for an expected interaction effect. The introduction needs to convey more clearly why this research is relevant and why this survey was conducted. It also needs to get clearer why there was a focus on the younger population and why there is any advantage of having the three concepts in one survey (same as why there would be an interaction effect)? I would like to ask the authors to rewrite the introduction to improve clarity and focus at the moment the only argument is that no one has done it before – which doesn’t prove the relevance of the question.

Response:

We have significantly reorganized the introduction, positing anxiety as a common factor for each of the three conditions being surveyed, noting that anxiety an important factor underlying each condition and is common in the young adults being studied. We have also tried to more clearly phrase our investigative hypothesis.

Results. In line with standard reporting of statistical results (see Bakker 2011, Behavior research methods) please can I ask the authors to report the statistical values, including t-values, chi-square values, and exact p-values for all comparisons that are the described throughout the manuscript and tables.

Response:

We have adopted the detailed APA guidelines to report the results of the Chi2 statistic and the t-test statistic. Results were reported for Chi2 as X2 (degrees of freedom, N = sample size) = chi-square statistic value, p = p value and for t-test as t(degrees of freedom) = the t statistic, p = p value.

Results are reflected both in-text and in appropriate Tables.

The abstract was not changed as per MDPI guidelines. 

Please can the authors explain the reasoning behind conduction person correlations AND multiple regression analysis? It should be clear from the outset which analysis will be used and why.

Response:

Thank you for this question, we have explicitly explained in the methodology. 

Correlation is a single statistic, or data point, whereas regression is the entire equation with all of the data points that are represented with a line after adjusting for covariates. Thus, correlation shows the relationship between the two variables, while regression allows us to see how one affects the other [42].

Please can the authors also clearly distinguish between meeting criteria on a questionnaire and an actual diagnosis of a disorder.

Response:

Yes, we agreed with this - We have clarified that – “All questionnaires used in this research are based on symptoms not on a diagnosis; thus, results need to be interpreted from that perspective. The issue of diagnosis versus symptoms is not only relevant to nomophobia but also to insomnia and food addiction. The insomnia questionnaire can provide a catalogue of symptoms suggestive of a diagnosis of insomnia, but is not the same as a clinical diagnosis of insomnia. Similarly, the food addiction questionnaire can provide an index of symptoms of individuals ‘at risk’ of food addiction; however, clinical diagnosis can be only made via an interview with a qualified healthcare professional.” This was added at the end of the discussion before making conclusions.

Discussion. The discussion reads at parts more like an introduction. Especially P6.L225 – P7.L244 this section lists information that should have already been introduced in the introduction. Furthermore, the discussion is not focused, instead it reads very listwise with future direction that, again, are listed but not expanded upon. It needs to be clear which research studies are needed to investigate the relationships found in the results and why. And it needs more than just a statement saying more research is needed. E.g. what kind of studies are the authors proposing?

Response:

The discussion was improved as per suggested directions. Parts that appeared to be literature review were modified into a discussion of related and similar findings. We also provided explanations of the different results. Finally, we have given practical suggestions to the kind of studies needed to advance knowledge.

I have a few minor corrections to propose. However, they will make more sense once the points above are addressed.

Reviewer 3 Report

Overall, the article is very well written, scientifically sound, and easy to follow. The authors may consider inserting a visual diagram showing the correlation (with r-value, p-value) between nomophobia and insomnia, as well as between nomophobia and food addiction. I do not have any major concerns. Thank you.

Additional comments:

In their article “The Association Between Nomophobia, Insomnia and Food Addiction Among Young Adults: Findings of a Cross-Sectional Survey”, the authors tested the association between nomophobia and insomnia, as well as between nomophobia and food addiction, in young adults (46% male, 54% female) using an online self-administered structured questionnaire. They found
that although there was no association between nomophobia and food addiction, there was a statistically significant association between nomophobia and insomnia (p=0.001, table 3).

The article is very well written and is easy to follow. The introduction has nicely described the background leading to the rationale for undertaking this study. Although the study period was short, it has been done on a scientifically sound design. The inclusion and exclusion criteria have been clearly described in the article. The methods have been described scientifically and explained as required. The study found that (1) moderate to severe nomophobia was very
common in the study population with prevalence of 93%, (2) prevalence of food addiction was very low at 19%, and (3) prevalence of moderate to severe insomnia was also very low at 14%.
The authors also found that that females were more affected with nomophobia, food addiction and insomnia in the study population.

Although overall I don’t have any major concern, I think the article might benefit from including visual diagrams showing the correlation graphs between nomophobia and food addiction, as well as between nomophobia and insomnia, if possible, along with their correlation coefficient values and p-values, which may make it more reader-friendly. 

Author Response

Overall, the article is very well written, scientifically sound, and easy to follow.

Response:

We thank the reviewer for taking the time to review our paper, and for the nice comments.

The authors may consider inserting a visual diagram showing the correlation (with r-value, p-value) between nomophobia and insomnia, as well as between nomophobia and food addiction. I do not have any major concerns. Thank you.

Response:

We have added a correlogram to help readers visualize the data in correlation matrices. All relationships in one graph. This correlogram summarize all relationships.

Additional comments:

In their article “The Association Between Nomophobia, Insomnia and Food Addiction Among Young Adults: Findings of a Cross-Sectional Survey”, the authors tested the association between nomophobia and insomnia, as well as between nomophobia and food addiction, in young adults (46% male, 54% female) using an online self-administered structured questionnaire. They found
that although there was no association between nomophobia and food addiction, there was a statistically significant association between nomophobia and insomnia (p=0.001, table 3). The article is very well written and is easy to follow. The introduction has nicely described the background leading to the rationale for undertaking this study. Although the study period was short, it has been done on a scientifically sound design. The inclusion and exclusion criteria have been clearly described in the article. The methods have been described scientifically and explained as required. The study found that (1) moderate to severe nomophobia was very
common in the study population with prevalence of 93%, (2) prevalence of food addiction was very low at 19%, and (3) prevalence of moderate to severe insomnia was also very low at 14%.
The authors also found that that females were more affected with nomophobia, food addiction and insomnia in the study population.

Response:

We thank the reviewer for the detailed summary , and for the nice comments.

Although overall I don’t have any major concern, I think the article might benefit from including visual diagrams showing the correlation graphs between nomophobia and food addiction, as well as between nomophobia and insomnia, if possible, along with their correlation coefficient values and p-values, which may make it more reader-friendly. 

Response:

Summary diagram was added in addition to the available correlation coefficient values and p-values

Round 2

Reviewer 2 Report

Dear Authors,

thank you for addressing my previous comments. I have the following suggestions:

  1. Please refer to insomnia symptoms, rather than insomnia, throughout (e.g. abstract/discussion)
  2. Please specify the population/age (Introduction, p2,L.59)
  3. Please check the English grammar and spelling throughout (e.g.p2,L.70)
  4. Introduction: it is still not clear to me why the authors decided to investigate insomnia and disordered eating? Why not any of the other comorbid problems that the authors describe? This decision still seems very random to me. Just because no one has looked at it in one population doesn't justify doing research. The additional information about anxiety as a common factor only makes me wonder why then anxiety wasn't measured. I think the authors need to be upfront - was this a predefined hypothesis or exploratory analysis? Was anything else measured other than the variables the authors describe?